# Dr. Google: Physicians—The Web—Patients Triangle: Digital Skills and Attitudes towards e-Health Solutions among Physicians in South Eastern Poland—A Cross-Sectional Study in a Pre-COVID-19 Era

**DOI:** 10.3390/ijerph20020978

**Published:** 2023-01-05

**Authors:** Joanna Burzyńska, Anna Bartosiewicz, Paweł Januszewicz

**Affiliations:** Institute of Health Sciences, Medical College of Rzeszow University, 35-959 Rzeszów, Poland

**Keywords:** online health information, digital literacy, e-health, e-health solutions, Dr. Google

## Abstract

The investment in digital e-health services is a priority direction in the development of global healthcare systems. While people are increasingly using the Web for health information, it is not entirely clear what physicians’ attitudes are towards digital transformation, as well as the acceptance of new technologies in healthcare. The aim of this cross-sectional survey study was to investigate physicians’ self-digital skills and their opinions on obtaining online health knowledge by patients, as well as the recognition of physicians’ attitudes towards e-health solutions. Principal component analysis (PCA) was performed to emerge the variables from self-designed questionnaire and cross-sectional analysis, comparing descriptive statistics and correlations for dependent variables using the one-way ANOVA (*F*-test). A total of 307 physicians participated in the study, reported as using the internet mainly several times a day (66.8%). Most participants (70.4%) were familiar with new technologies and rated their e-health literacy high, although 84.0% reported the need for additional training in this field and reported a need to introduce a larger number of subjects shaping digital skills (75.9%). 53.4% of physicians perceived Internet-sourced information as sometimes reliable and, in general, assessed the effects of its use by their patients negatively (41.7%). Digital skills increased significantly with frequency of internet use (*F* = 13.167; *p* = 0.0001) and decreased with physicians’ age and the need for training. Those who claimed that patients often experienced health benefits from online health showed higher digital skills (−1.06). Physicians most often recommended their patients to obtain laboratory test results online (32.2%) and to arrange medical appointments via the Internet (27.0%). Along with the deterioration of physicians’ digital skills, the recommendation of e-health solutions decreased (*r* = 0.413) and lowered the assessment of e-health solutions for the patient (*r* = 0.449). Physicians perceive digitization as a sign of the times and frequently use its tools in daily practice. The evaluation of Dr. Google’s phenomenon and online health is directly related to their own e-health literacy skills, but there is still a need for practical training to deal with the digital revolution.

## 1. Introduction

By the beginning of 2022, the number of internet users had reached a global 4.95 billion, which means 65% of the world population is online [1]. The Internet has undoubtedly become one of the most popular sources of medical information in the field for the definition, symptoms, and diagnosis of diseases, as well as healthy lifestyles, medications, alternative medicine, medical devices, and the functioning of the entire medical care system [2,3,4,5]. Studies confirm that searching for health-related content is not the only reason to use Internet—consumers more and more frequently before seeing a doctor ask Dr. Google for online consultations [3,6,7,8,9]. Researchers pointed out that people who seek health materials on the Web differ in age, income, and education levels from those seeking information offline [8]. However, searching for digital information on diseases is not only using search engines. More and more websites are being created. Virtual symptoms checkers exist, where users can quickly (in about 5 min), free of charge, and without registration identify possible conditions after entering the symptoms and providing the answer to additional questions, thus making for an initial diagnosis. There are data indicating that younger people who are in worse health condition seek access to this information more often [10]. They are also the ones with a higher socio-economic status and are better educated, most often women [11]. Interestingly, half of all searches are made on behalf of another person [11]. The high percentage of the above indications was determined by the quick access to information guaranteed, the ease of obtaining it, and the possibility of better preparation for a doctor’s visit [9]. The offer of the global network in the health area is plenteous: from pharmacies and clinics search engines, through expert advice, discussion forums, blogs of patients, and professionals, virtual support groups, tests, symptom checkers, and ending with websites devoted to specific health-related issues (e.g., diet, activity, prevention), diseases, and drugs.

This situation begins to occupy an equal place in the doctor–patient interaction. Previous studies suggested that the relationship between consumers and healthcare professionals may be a motivation to search for health information online [2,12,13]. Research indicates that Internet-based decision-making, independent of the physician, can improve patients’ self-efficacy and reduce unnecessary visits to physicians [14]. On the other hand, doctors are particularly critical in the process of implementing digital health in the healthcare sector. They mainly report a lack of proper preparation and training [15] and difficulties in adapting to the rapidly progressing changes in this area. Some initiatives are known to provide specialization training on medical informatics, such as Topol review [16], but they have not been developed in all countries. At the same time, the situation of clinicians who are aware that their patients seek the popular Web domains for health information was examined [17,18,19]. This phenomenon, known as “Dr. Google”, identifies the complex effects and consequences of searching for online health information—a kind of medical autonomy for patients, as well as the role of professionals with their appropriate knowledge and practice [14]. 

Despite digital health popularity, the literature suggests that patients still value the health professionals’ advice [14,19], which might be a suggestion that physicians can play an important role in online health navigation, and the need to have a proficiency in digital skills is even more vital. The implementation of information communication technology (ICT) may be the answer to this challenge and can provoke us to remain open to technological opportunities and the possession of skills to manage them [20]. For physicians, ICT competences might be relevant for several reasons, starting with the fact that medical technology is advancing and daily tasks handling health data is challenging (e.g., electronic health records, electronic documentation, and health information systems) and ending with patient–consumer personalized and individualized demands [21]. However, today’s profession practice is not only dependent on this duo. There is ample evidence that technology solutions increase the efficiency of health services and improve patient outcomes. Electronic medical records (EMR) have been shown to significantly reduce the incidence of medical errors, and e-prescriptions have been shown to improve prescribing errors and patient adherence to treatment regimens [22]. Moreover, there is evidence that e-Health systems have led to fewer hospital visits and cost savings in the healthcare system [23]. 

In the new European Skills Agenda, the European Commission sets objectives for upskilling (improving existing skills) and reskilling (training in new skills) the workforce in the next five years [24]. In its policy paper on digital skills for doctors, the Standing Committee of European Doctors highlights the three main areas of digital skills: general, technical, and related to patient–doctor relationship [25]. These new competences should be included in the future medical programs of study and adapted to a doctor’s medical speciality, and they should keep current and future physicians up to date with tech innovations.

Few years ago, the European Commission defined resources to efficiently use the possibilities of new technology as digital literacy [26]. According to Bawden, digitally literate people have the knowledge and ability to effectively and critically navigate, evaluate, and create information using a range of digital technologies, as well as using Internet and technology tools to achieve academic, professional, and personal goals [27]. The medical education of Polish physicians is evidence-based, but, in the context of the challenges related to Dr. Google, technological development, and the ubiquity of the Internet, it might not be sufficient. Health technologies provide broad access to resources, enable easy search for information, and support initiatives related to global health care. These technologies have become a trend, transforming healthcare systems in the context of creating ones that are largely open to all patient needs. Due to the development of such technologies, the need for digitally trained personnel arose. 

## 2. Theoretical Background and Literature Review

### 2.1. Physicians’ Attitude and Acceptance of e-Health

The increase in the number of users of health websites is proportional to the global increase in the number of people using the Internet. In 2000, an initiative was taken to enable the citizens of the European Union to provide new generation medical services by supplementing the “e-Europe 2002” document with the thematic area of e-health, which includes the above facilities. The term e-health is defined by the European Commission as: “the use of modern ICT technologies to meet the needs of citizens, patients, healthcare professionals and health service providers” [28]. Thanks to e-health services, healthcare more precisely adjusts to the individual needs of patients and, as a result, makes the treatment process more effective [29]. Patients are not passive in this matter, using the practical dimension of e-health, which is, e.g., access to health information via the Internet. The results of the 2007 international project commissioned by WHO (World Health Organization), entitled The European e-Health Consumer Trends Survey, in order to determine the trend of consumer needs in the field of e-health, showed that 52.2% of Internet users used it to search for such information [30]. Health is a topic that also interests Polish society. As the report “e-Health through the eyes of Poles” indicates, Polish society has high hopes for the dynamic development of electronic solutions [31]. Research shows that, by far, the largest percentage of respondents count on the reduction of queues to specialists, along with the development of medical technologies (60%), and at the same time, they hope that a medical doctor will devote much more time to patients than “paper” duties (58.7%) [32]. 

When it comes to physicians’ attitude towards health technology, e-health tools were well-received, but the adoption of new technology was very slow [33]. As the main barriers to the implementation and adoption of new technologies, professionals mainly focused on: the lack of harmonization of e-health systems [34] and usability issues, such as the ease of use and intuitiveness of the solutions [35]. Studies have also cited the lack of time and workload experienced by physicians as other key barriers to the implementation of e-health technology [36]. Privacy and security issues are also a barrier reported by the surveyed physicians [37]. An important aspect also indicated by many practitioners was the lack of IT skills, which may be perceived as a barrier to the use of e-health technologies [38]. Consequently, the need for training and support is also highly reported as an important facilitator to the adoption of e-health technology [39].

The mentioned annexation of various areas of human functioning by information and communication technologies forces competing producers to conduct more and more in-depth analysis of consumer tastes and preferences. In particular, with regard to new health technologies, where the long-term effects of their use are difficult to predict, detailed analyses are necessary. Analytical tools that work in many areas of life, including health care, are theories and models of technology acceptance, such as TAM (technology acceptance model) or UTAUT (Unified Theory of Acceptance and Use of Technology). Both models have found a widespread adoption in healthcare for investigating the factors affecting users’ acceptance of novel technical systems [40]. The technology acceptance model (TAM), originally proposed by Davis [41], is one of the most widely used theoretical models for predicting and explaining whether users will accept the development and application of new IT or other systems. The theory posits that a person’s intention to use and usage behavior of a technology are predicated by the person’s perceptions of the specific technology’s usefulness (benefits from using the technology) and ease of use [42]. UTAUT adds two further factors associated with the social impact and technical infrastructure to use the digital technology. TAM hypothesizes that two particular beliefs, i.e., perceived usefulness and perceived ease of use, are the primary relevance for technology acceptance [43]. In UTAUT, the behavioral intention to use the technology is determined by three factors: performance expectancy, effort expectancy, and social impact [42,44]. In empirical tests, the UTAUT model could explain approximately 70% of the variance in a user’s behavioral intention to use a new technologies [45]. TAM has been also applied to investigate the factors affecting healthcare professionals’ acceptance of e-health technology [38], and inconsistent results have been found. Yarbrough and Smith [46] argued that one of the limitations of TAM is its inability to account for the influence of external variables and barriers to technology acceptance.

### 2.2. Research Questions

In the 2020s, it seems impossible not to pay attention to the impact of technology on healthcare. However, despite the large investments in digital technology, the evidence of the impact of its benefits on health systems is still not fully discovered. The aim of the current study is to understand the digital aspects of healthcare from the perspective of Polish physicians. We proposed the following research questions:Q.1 How do physicians evaluate their digital skills and digital literacy, and what variables determine this?Q.2. How do physicians perceive the quality and usefulness of online health information for their patients, and what variables determine this?Q.3. How do physicians evaluate e-health solutions in general, and what variables determine this?Q.4. Do physicians recommend e-health solutions to their patients, and what variables determine this?

This study also aims to reflect the actual state of digitization and identify the dominant variables and the most frequently used factors to fill the gaps in the literature and help further research to build an integrated strategy or models of digital relevance in the field of healthcare. To our knowledge, no research exists examining digital literacy, perception of online health information, and attitudes towards e-health solutions from physicians’ perspective in Poland. As a part of a larger study of healthcare professionals and patients, the aim of this research was to better understand the phenomenon, in the aspect of digitalization of Polish healthcare sector. Physicians’ opinions of self-digital skills, health internet-source information, and e-health solutions also seem particularly interesting in the context of the digitization of medical services.

## 3. Materials and Methods

### 3.1. Study Design

This is a cross-sectional, descriptive study conducted among 307 professionally active physicians between December 2019 to April 2020, in the South-Eastern Poland. The sample was representative of the group of doctors, which consisted of 4600 specialists in 2018 in the Podkarpackie Voivodeship. Additionally, the number of doctors living in Poland with the right to practice in 2019 amounted to 140,420, of which 58.9% were women. The most common specialization of Polish doctors was internal medicine (20.9%), the second largest was family medicine (11.8%), the third—paediatrics (8.2%) [47]. The current study is a part of a larger ongoing project conducted to explore attitudes towards e-health, digital literacy, and online health information among healthcare sector representatives (physicians, nurses [48]) and patients. The self-designed questionnaire used in this research was distributed among the following healthcare facilities employing physicians: primary healthcare (PHC), outpatient specialist care (AOS), hospitals (at different levels of locality), and private sector. Respondents participated in this study ensured sample diversity working in southeastern Poland (the error threshold was 5%, i.e., the test power was 0.95). Before participating in the survey, respondents were provided with detailed information about the purpose of the study, assured of its anonymity, and that they have the option to opt out at any stage of it.

### 3.2. Questionnaires

This survey adopted a structured self-designed questionnaire to obtain data from physicians and included a sociodemographic section with questions about age, gender, specialization, workplace, and number of patients seen per month. Surveyed physicians were also asked to indicate the frequency of using the Internet and electronic devices.

The questionnaire had also four sections that included:“Digital Literacy” section on self-assessment of the ability to use digital devices and solutions, and evaluation the frequency of using digital devices and solutions in private and professional life (smartphone, computer, e-mail, mobile applications, and tablet);“The impact of the Internet/new technologies on healthcare and modern life” section concerning the general assessment and opinion on the use of the search engine/Internet in the context of health;“Recommendation of e-Health solutions” section containing statements regarding the present/future recommending or not recommending e-health solutions to the patients,“Evaluation of e-Health solutions” section with self-assessment of the relevance of the proposed e-health solutions;

Questionnaire template is included in Appendix A.

### 3.3. Ethics

The study was approved by the institutional Bioethics Committee of the Rzeszow University (Resolution No. 6/12/2019) and all relevant administrative bodies.

### 3.4. Data Analysis

In this study principal component analysis (PCA), using the Anderson–Rubin method, was carried out to determine constructs in questionnaire regarding: “Digital literacy”, “The impact of the Internet/new technologies on healthcare and modern life”, “Recommendation of e-Health solutions”, and “Evaluation of e-Health solutions”. The reliability of each construct was also determined by the Cronbach’s alpha coefficient. A significance level of *p* < 0.05 was assumed.

Descriptive statistics were conducted to present the data: frequency (n), percentage (%), arithmetic mean (M), the value of which determines the average level of a given variable, and standard deviation (SD), a statistical measure of scattering the results around the expected value.

Then, correlation analyses were performed. The t-tests and chi-square tests for two-group comparisons in sociodemographic variables (gender and workplace) were performed. Nonparametric Spearman Rho correlation coefficient test was used to compare the non-normally distributed numerical data in correlation analysis with age. Differences in mean values in the cross-sectional analysis were calculated using the one-way analysis of variance for the cross-sectional groups ANOVA (*F* test). To measure the linear relationship between all digital indicators, the Pearson correlation (*r*) was used. 

Statistical significance was evaluated at *p* < 0.05.

Calculations were performed with the IBM SPSS program Statistics 20 (IMB, Armonk, NY, USA).

## 4. Results

### 4.1. The Reliability and Validity of Questionnaires

The Cronbach’s alpha and PCA were calculated for each construct in questionnaire:Digital literacy: the Cronbach’s alpha: 0.571, PCA: two components (“Own skills”, and “Need for training”), explaining 61.6% of the variance;The impact of the Internet/new technologies on healthcare and modern life: the Cronbach’s alpha: 0.462, PCA: one variable explaining 65.0% of the variance;Recommendation of e-health solutions: the Cronbach’s alpha: 0.865, PCA: one component explaining 52.1% of the variance;Evaluation of e-health solutions: the Cronbach’s alpha: 0.928, PCA: two components (“Patients” and “Medical facility”), explaining 69.0% of the variance.

### 4.2. Characteristics of the Study Group

A total of 307 physicians participated in the study, including 168 (54.7%) females and 139 (45.3%) males. The mean age of the respondents was 44.39 ± 14.21 years, ranging from 26 to 78 years. Every fourth physician was not more than 31 years old, and every fourth was not younger than 54 years old. The dominant specialization (N = 136; 44.3%) among practitioners was family medicine. A total of 32 (10.4%) paediatricians, 28 (9.1%) orthopaedists, 27 (8.8%) internists, 18 (5.9%) neurologists, and 11 (3.6%) gynaecologists also participated in the study. Data were also collected from 7 (2.3%) cardiologists and anaesthesiologists, 6 (2.0%) pulmonologists, 5 (1.6%) dermatologists, 3 (1.0%) dentist, 6 (2.0%) residents and 9 (2.9%) doctors during specialization. The vast majority of participants were employees of primary healthcare (81.4%), less often employees of hospitals (34.9%) or other places. A total of 253 (82.4%) of physicians also conduct private practice. Nearly half of professionals (N = 148; 48.2%) saw more than 200 patients within a month. Between 50 and 100 patients were seen monthly by 28% (N = 86) of physicians, and 12.4% (N = 38) declared monthly care for less than 50 patients. 11,4% (N = 35) professionals saw monthly between 100 and 200 patients (Appendix A). The surveyed clinicians reported they use the Internet mainly several times a day (66.8%) or every day (19.9%) (Figure 1) (Appendix A).

In private life, physicians mainly used a smartphone (83.4%), e-mail (70.7%), and a computer (65.8%). In their professional work, clinicians mainly use a computer (85.0% often) and smartphone (47.6% often) (Table 1).

### 4.3. Physicians’ Digital Literacy and e-Health Indicators

Clinicians described their skills in using digital devices or e-health solutions as very good (33.6%) or good (32.6%) (Appendix A). However, most of the respondents admitted they will take advantage of additional training or courses in the field of shaping digital competences (84.0%), and there is a need to introduce a larger number of subjects shaping digital skills (75.9%) in medical studies, but they felt prepared to support e-health solutions (70.4%). Clinicians disagreed with the statement (43.0%) that today’s medical education keeps pace with the digital challenges of the 21st century. Participants shared more (40.1%) or less (37.8%) the opinion that the Internet would revolutionize healthcare, and in 57.3%, they considered new technologies helpful in the modern life (Table 2).

To verify how the frequency of using the Internet and the number of patients seen on average per month affect digital literacy, as well as other e-health variables, the appropriate analyses have been carried out. It was shown that the frequency of using the Internet had a significant impact on the discussed indicators. Respondents who used the Internet every day (−0.22) or several times a day (−0.09) had higher digital skills (*F* = 13.167; *p* = 0.0001). Physicians using the Internet several times a day indicated a greater need for digital training (0.20). It was also shown that the more frequent the use of the Internet, the higher the assessment of the impact of the Internet and new technologies on healthcare and modern life (*F* = 11.448; *p* = 0.0001). A higher level of recommendation of e-health solutions concerned physicians using the Internet several times a day (−0.17) than other respondents, using the Internet less frequently (*F* = 7.045; *p* = 0.0001). E-health solutions in relation to the patient were also rated higher (*F* = 37.800; *p* = 0.000) by physicians using the Internet several times a day (−0.25) or daily (−0.06). It was noticed that the evaluation of e-health solutions, in relation to the medical facility, was higher (*F* = 3.343; *p* = 0.0107) in respondents who used the Internet once a week (−0.79) or did not use it at all (−0.22). The impact of the Internet and new technologies on healthcare and the modern life was rated higher (*F* = 5.371; *p* = 0.0013) by physicians who saw less than 50 patients per month (−0.35), and with the number of patients admitted that the assessment of the impact of the Internet on healthcare decreased. It was also noticed that professionals who have seen less than 50 patients per month recommended more (*F* = 5.432; *p* = 0.0012) e-health solutions (−0.34), and rated e-health solutions higher (*F* = 5.163; *p* = 0.0017), in the relation to a medical facility (−0.42) (Table 3). 

It was also checked how the frequency of using selected devices (in private life and at work) affects digital literacy, impact assessment, recommendation, and evaluation of e-health solutions by physicians. It has been noticed that there were statistically significant differences between the use of a computer in private and professional life and digital literacy (both skills and training needs). Respondents who frequently use the computer in their private life had a higher index of digital skills (*F* = 9.480; *p* = 0.0000). A similar relation was made with the use of a computer at work (*F* = 17.886; *p* = 0.0000). Physicians who used the computer less frequently in their private life or work had lower digital literacy, in terms of skills (obtained higher scores, therefore their skills were lowered). Another statistically significant relation has been recognized, indicating that respondents who often used a tablet, smartphone, e-mail, and mobile applications in their private or professional life had higher digital literacy, in terms of skills. The impact of the frequency of computer use on digital literacy, assessment of the Internet impact, recommendation of e-health solutions, and assessment of these solutions (in relation to the patient) was also statistically significant. Respondents who frequently use the computer (both in private and professional life) also saw a greater need for training in the field of digital literacy, rated the impact of the Internet and new technologies on healthcare and a modern life higher, recommended e-health solutions more often, and rated higher e-health solutions, in relation to the patient (Table 4). The above results answer the research question Q.1. (More analyses confirming the above results can be found in the Appendix A).

There were no significant differences between gender and the values of selected e-health indicators; however, it was shown that, with the age of surveyed physicians, digital skills decreased significantly, as well as the need for training. Moreover, the older the respondent, the lower the assessment of the impact of the Internet and new technologies on healthcare and modern life, the recommendations of e-health solutions, and the evaluation of e-health solutions, in relation to the patient. With physicians’ age, the assessment of e-health solutions related to running a medical facility increased (Appendix A).

### 4.4. Physicians’ Opinion about Online Health Information (Dr. Google)

Every third physician (33.2%) acknowledged that more than 10% of her/his patients access health information on the Internet within a month. In their opinion, such information was most often perceived as sometimes reliable (53.4%). A total of 34.5% of physicians admitted that their patients sometimes experienced health benefits from accessing material on the Internet, and 23.8% considered it to be rare. Generally, clinicians negatively assessed (41.7%) the effects of using Internet-based health knowledge by their patients, which partially answers research question Q.2 (Appendix A).

### 4.5. Dr. Google and Physicians’ Digital Literacy Indicators

A greater need for training (lower digital literacy in this area) was demonstrated by physicians who stated that less than 1% of their patients access online health information (0.55). At the same time, these clinicians rated e-health solutions higher than others (−0.38), in relation to the patient. The generally reliable online health information was assessed mainly by physicians who rated the impact of the Internet and new technologies higher (−0.82), recommended e-health solutions (−1.52), and rated e-health solutions higher, as related to the patient (−0.85). Respondents who did not have an opinion on the quality of online health information were less interested in digital training (−0.44) and rated e-health solutions higher, in relation to a medical facility (−0.53). Physicians who claimed that patients often experienced health benefits from accessing online health information showed higher digital skills (−1.06), rated the impact of the Internet on healthcare higher (−0.50), recommended e-health solutions more (−1.53), and rated e-health solutions higher (−0.84). Respondents who said that their patients had never experienced health benefits from online health information had a higher need for digital training (0.98). Physicians who very positively assessed the effects of patients’ use of online health knowledge had greater digital skills (−0.62). Those who assessed the effects negatively were the least likely to have digital training. Respondents who very positively assessed the effects of patients using online health knowledge at the same time rated the impact of the Internet on healthcare higher (−0.52), recommended e-health solutions (−1.52), and rated e-health solutions higher, in relation to the patient (−0.72). Physicians who negatively determined the effects of patients using online health knowledge at the same time rated e-health solutions, in relation to a medical facility (0.21) (Table 5). (More analyses between type of workplace can be found in Appendix A.

The percentage of patients accessing online health information within a month was a variable that significantly correlated with the need for training in physicians’ digital literacy and the assessment of e-health solutions, in relation to the patient. The physicians’ opinion about the overall quality of online health information correlated significantly with all other variables related to digital literacy and e-health indicators, as well as the opinion on patients’ health benefits of online materials did, with one exception, regarding the assessment of e-health solutions, in relation to medical facilities. On the other hand, the assessment of the effects of using the Internet-based health knowledge by patients fully correlated with all the indicated variables (Table 6). The above results answer research question Q.2.

### 4.6. Recommendation of e-Health Solutions

Physicians most often recommended that their patients obtain laboratory test results (32.2%) and arrange medical appointments via the Internet (27.0%). The willingness to recommend was mainly (72.3%) related to the use of mobile applications reminding about the need to take medications. Doctors did not recommend applications facilitating the analysis of test results (48.9%), did not recommend video consultation (46.3%), and were not in favour of the remote monitoring of vital parameters (44.0%) (Table 7). The above results answer the research question Q.3.

### 4.7. Evaluation of e-Health Solutions

The most important e-health solutions considered by physicians were: using the electronic database of medicines (48.2%), solutions facilitating the sending/exchange of clinical results (46.6%), the possibility of writing out electronic sick leaves (44.6%), as well as easy and quick access to the patient’s medical records in electronic form (42.0%) (Table 8). The above results answer the research question Q.4.

### 4.8. Correlation Matrix of All Digital Indicators

It has been shown that, along with the deterioration of digital skills, the assessment of the impact of the Internet and new technologies on healthcare decreased (*r* = 0.447; *p* = 0.000), as well as the recommendation of e-health solutions (*r* = 0.413; *p* = 0.000) and the assessment of e-health solutions for the patient (*r* = 0.449; *p* = 0.000). The lower the need for training in digital literacy, the higher the assessment of the impact of the Internet and new technologies on healthcare (*r* = −0.239; *p* = 0.000), the recommendation of e-health solutions (*r* = −0.131; *p* = 0.000), and the assessment of e-health solutions, in relation to the patient (*r* = −0.250; *p* = 0.000). The lower the assessment of the impact of the Internet and new technologies on healthcare, the lower the recommendation of e-health solutions (*r* = 0.369; *p* = 0.000) and the rating of e-health solutions (*r* = 0.457; *p* = 0.000). It was also found that, the lower the recommendation of e-health solutions, the lower the evaluation of e-health solutions for the patient (*r* = 0.588; *p* = 0.000) and for the medical facility (*r* = 0.140; *p* = 0.000) (Table 9).

## 5. Discussion

We live in a world of rapid technological, demographic, and social changes, in which digitization has become a factor of advantage and an element of the competitiveness of economies and health systems, as well as health policies. These changes are becoming a fact that needs to be documented in various aspects: technological, human, qualitative, financial, etc. Especially in the case of Poland and many other countries in the region, the development and popularization of digital e-health services is a priority direction for the development of the local health care system [49]. 

Our purpose in this study was to investigate physicians’ self-digital literacy, analyze their perception of online health knowledge obtained by patients, and identify attitudes towards e-health solutions. It is also important to characterize and describe the phenomenon of Dr. Google, which affects the existing doctors–patients’ relationships, and shapes the modern understanding of health information. 

In general, the majority of 307 physicians who participated in the study were familiar with new technologies. Both in their private and professional life, they use a smartphone (83.4% vs. 47.6%), a computer (65.8% vs. 85.0%), and e-mail (70.7 % vs. 30.9%). Respondents reported using the Internet mainly several times a day (66.8%) or every day (19.9%), which does not differ significantly from the statistics in previous studies [2,8,21,47,50]. In the “Internet in the life of Poles” report commissioned by the Ministry of Digitization in June 2019, the professional group most willing to use the Internet and e-services were physicians [51]. 

A similar situation applied to the skills to use e-health—clinicians described them as very good (33.6%) or good (32.6%). Moreover, our study has demonstrated that the more intensive the Internet use, the higher digital skills (*F* = 13.167; *p* < 0.0001). However, according to the CPME Policy on Digital Competencies for Doctors, neither the practicing health professionals nor the generation in training are adequately prepared, and there is still a gap between understanding how digital solutions may support their capabilities, as well as the actual use of them [25]. The self-reported level of digital skills in this study reflects professionals’ confidence and understanding the status new technologies in healthcare.

In this context, interesting results were obtained: although 70% of respondents felt prepared to support e-health solutions, still a significant percentage (84%) reported the need for additional training or courses in the field of shaping digital skills. Additionally, the more the use of electronic devices (computer, tablet, smartphone, e-mail, and mobile apps), the greater the need for training in the field of digital literacy that was reported. The study prepared for the European Parliament confirms these fears: of the participants that received digital skills training, 54% rated it as insufficient. A survey of more than 200 physicians found that a large majority (80%) of the questioned professionals indicated that the currently available e-health/m-health training is inadequate [52]. Especially since digital skills have been shown to deteriorate with the age of surveyed physicians. These are important findings, according to the dominant opinion (77.9%) among physicians that the Internet would revolutionize modern healthcare. The awareness of the importance of digitization, in the case of the surveyed doctors, was connected with the explicit doubt regarding whether the educational programs in medical studies follow the technological challenges of the 21st century. It is a paradox that, in times of an unprecedented technological boom, medical education in this field is still insufficient, as multiplicity studies from different countries have reported [53,54,55].

The use of the Internet for health purposes is increasing, but its impact on healthcare is still unclear. Undeniably seeking health information online will keep increasing [56]. People indicate the Internet as the first source of health information primarily because of limited time consultation and barriers to accessing professional health services [57]. However, how the above translates into interaction: physician-the Web-patient? Cocco et al. found that seeking online health information by patients had positive effects on the doctor–patient relationship and improved their communication. However, at the same time, it was noticed, that the Internet-sourced information did cause some anxiety for 40% of searchers [58].

Polish Internet users most often (almost 59%) pointed that data they can find on the Internet does not require leaving home. Almost equally important (53%) was the opportunity to find the information 24/7. Another reason indicated by 32% of respondents was to supplement the information that they did not receive during a medical visit. Thanks to the information obtained earlier, almost 25% of respondents felt better prepared to visit a doctor. The next places were: the possibility of finding information that is not elsewhere (23%), anonymity (20%), or the ability to contact other people in a similar situation (16%) [59]. Based on the national survey conducted in 2013 by the Pew Research Center’s Internet and American Life Project, 1 in 3 American adults said they went online to determine their medical condition, and 35% of the respondents said they do not need any professional opinion [11]. In our study, every third physician (33.2%) acknowledged that more than 10% of her/his patients access health information on the Internet within a month. A total of 85% physicians questioned by Murray et al. had experienced a patient bringing Internet information to a visit [13]. The quality assessment of this information in physician’s opinion in our study was most often perceived as sometimes reliable (53.4%). Potts and Wyatt came to similar conclusions, reporting that over two thirds of the questioned professionals considered online health information to be usually (20%) or sometimes (48%) reliable [60]. Again, data from our study showed that physicians who assessed the effects of using the online health knowledge by patients very positively had greater digital skills, recommended e-health solutions, and rated e-health solutions higher. At the same time, those who assessed the effects of the use of the Internet by patients negatively indicated the need for training in digital literacy. We did not check whether these results significantly affect the doctors–patients’ relationship, but Murray et al. reported that physicians did have an impression that their authority was challenged because of Dr. Google [13]. Moreover, this reaction was associated with harms to the quality of care, health outcomes, and time efficiency. The results of other studies indicate that physicians’ attitude to online health information seeking is critical [43], and they are not very familiar or confident with patients who ‘Google’ [61].

In the context of recommending and assessing e-health solutions, Polish physicians remained favourable and expressed a definite desire to recommend such solutions to their patients. It is a limitation that our study was conducted before the strongest influence of COVID-19 pandemic, which further shows the positive attitude of doctors to this type of solutions. The pandemic has forced changes in digital health, which should be investigated, but in its source, modern health care refers to the educational role of a doctor who can have an active impact on the patient, providing reliable and trustworthy recommendations of new technological solutions dedicated to health [17,62]. However, as our research showed, with the lower digital skills of physician, the recommendation of e-health solutions decreased, and the assessment of e-health solutions for the patient was reduced.

To sum up, the strategic importance of digital skills goes beyond the issue of the short-term development of e-health itself—it should be considered as a key factor of the development of conscious information society, both on the side of the professionals and the recipients of services, neither in highly developed nor in developing countries. Smartphones, universal access to the Internet, and mobile applications meeting patients’ expectations are revolutionizing the sector. Additionally, telemedicine began filling into clinical practice, medical universities began including e-health in their curricula, and health systems are introducing legal regulations supporting digital health.

The issue of digitization in healthcare is often referred to as a process depending only on the technology to which people should adapt. Digital reality, however, is not only the e-development, but also the need for a holistic approach to the entire sector, organizational processes, legal regulations, and standards that guarantee interoperability and cooperation between the physician, the Web, and the patient.

## 6. Conclusions

This study provides some contribution to theory and practice. From the theoretical perspective, we explore the physicians’ digital literacy, their opinion about Dr. Google (a phenomenon understood as searching and obtaining information on health via the Internet), and overall assessment of e-health solutions. Our empirical study confirms that, despite the highly rated digital skills, doctors reported the need for further training in addressing digital competences. Our research led to the conclusion that professionals and authors of curricula for medical studies should cooperate to develop consistent courses that meet today’s digital challenges. Moreover, the results showed that physicians generally stay sceptical about online-sourced health knowledge obtained by their patients, but they do believe that Internet will revolutionize healthcare in the near future. Digital health technologies are changing the way healthcare is delivered; therefore, it seems that the educational potential and indicating to the patient which content dedicated to health is valuable and useful is one of the important tasks of a modern physician.

## 7. Limitations

This study had several limitations. First, the research relied on a self-designed questionnaire. At the stage of creating study, the available tools for measuring digital literacy should be included; however, in this case, we wanted to use specific sections in the questionnaire, and each construct was validated. Second, the study was conducted before the COVID-19 pandemic, which is important for the interpretation of results obtained. There is scientific evidence that this global pandemic significantly accelerated the daily use of digital solutions in the health system [63,64]. According to the Deloitte report, almost 65% of the Polish health system employees indicated that, in reaction to the COVID-19 pandemic, their institutions increased the use of digital technologies supporting physicians’ work, and in the opinion of 64.3% respondents, they were also used in remote support and contact with patients. At the same time, the use of digital solutions was most often mentioned by GPs (74.7%), who, due to the COVID-19, widely adopted the remote method of the initial patient assessment [64]. This could have a positive impact on the attitude of physicians and patients towards e-Health solutions and should be the subject of further, post-pandemic research. Another limitation we acknowledge is that we surveyed physicians from one region in Poland, which may have limited the generalizability to the country and beyond. However, we gathered a large and diverse group of physicians of various specialties, representative of the Podkarpackie Voivodeship. It would be useful to repeat our study in the whole country to determine the attitudes of a larger sample of professionals in more facilities and regions than we were able to achieve. Future research may also look at individual medical specialties for a deeper understanding of their perspectives. Finally, as the study is cross-sectional in nature, alternative relationships might also exist; thus, future research should be longitudinal.

## Figures and Tables

**Figure 1 ijerph-20-00978-f001:**
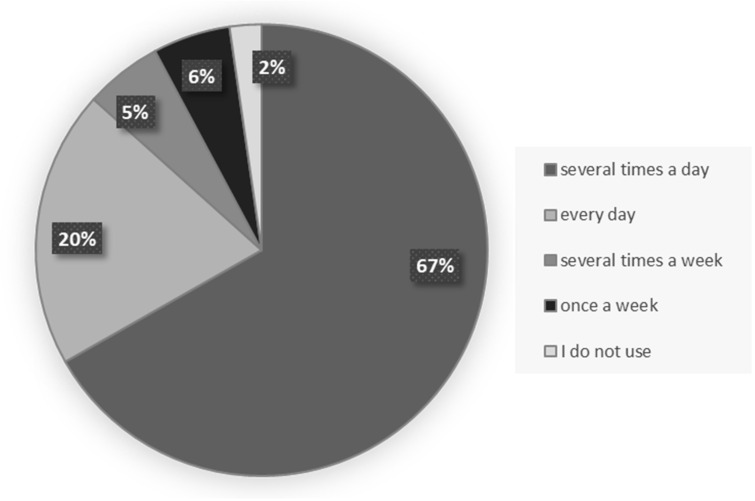
Distribution of the frequency of internet use by surveyed physicians.

**Table 1 ijerph-20-00978-t001:** The frequency of the electronic devices usage by physicians in private and professional life.

Type of Electronic Device	Often		Sometimes		Never		N/A	
	N	%	N	%	N	%	N	%
Private usage								
Smartphone	256	83.4	23	7.5	24	7.8	4	1.3
Computer	202	65.8	94	30.6	8	2.6	3	1.0
E-mail	217	70.7	59	19.2	20	6.5	11	3.6
Mobile apps	173	56.4	80	26.1	44	14.3	10	3.3
Tablet	70	22.8	104	33.9	107	34.9	26	8.5
At work								
Smartphone	146	47.6	75	24.4	65	21.2	21	6.8
Computer	261	85.0	31	10.1	14	4.6	1	0.3
E-mail	95	30.9	116	37.8	82	26.7	14	4.6
Mobile apps	76	24.8	103	33.6	103	33.6	25	8.1
Tablet	16	5.2	21	6.8	224	73.0	46	15.0

**Table 2 ijerph-20-00978-t002:** Physicians’ digital literacy and the aspects of this field.

Item	Yes		Rather Yes		No Opinion		Rather No		No	
	N	%	N	%	N	%	N	%	N	%
I feel prepared to use e-Health solutions in my work	77	25.1	139	45.3	40	13.0	48	15.6	3	1.0
I would use training courses to improve my digital literacy	97	31.6	161	52.4	28	9.1	20	6.5	1	0.3
Current physician education keeps pace with the digital challenges of the 21st century	45	14.7	50	16.3	80	26.1	116	37.8	16	5.2
There should be more subjects that shape digital literacy in medical studies	80	26.1	153	49.8	43	14.0	30	9.8	1	0.3
The Internet will revolutionize healthcare in the near future	123	40.1	116	37.8	52	16.9	12	3.9	4	1.3

**Table 3 ijerph-20-00978-t003:** Correlations between: the frequency of Internet use and the number of patients seen on average per month by physicians, as well as digital and e-Health selected variables.

	Digital Literacy—Own Skills		Digital Literacy—The Need for Training		The Impact of the Internet/New Technologies on Healthcare and Modern Life		Recommendation of e-Health Solutions		Evaluation of e-Health Solutions—The Patient		Evaluation of e-Health Solutions—Medical Facility	
	Mean	SD	Mean	SD	Mean	SD	Mean	SD	Mean	SD	Mean	SD
The frequency of Internet use												
several times a day	−0.09	0.93	0.20	0.91	−0.14	0.97	−0.17	0.92	−0.25	0.66	0.02	0.92
everyday	−0.22	0.76	−0.10	0.95	−0.07	0.93	0.17	1.05	−0.06	0.96	0.14	0.99
several times a week	0.11	0.60	−0.86	0.99	0.31	0.67	0.22	1.02	0.67	1.24	0.21	1.34
once a week	1.27	1.19	−0.58	1.14	0.91	0.54	0.88	0.89	1.66	1.08	−0.79	0.91
no use	1.36	1.83	−1.48	0.89	1.70	1.28	0.81	1.43	2.22	1.66	−0.22	1.76
	*F*	*p*	*F*	*p*	*F*	*p*	*F*	*p*	*F*	*p*	*F*	*p*
	13.167	0.0000	12.133	0.0000	11.448	0.0000	7.045	0.0000	37.800	0.0000	3.343	0.0107
The number of patients seen on average per month												
<50	0.01	1.09	−0.05	1.02	−0.35	0.89	−0.34	1.22	0.07	1.29	−0.42	0.65
50–100	−0.04	0.95	0.21	0.64	−0.23	0.85	0.28	0.75	−0.05	0.72	0.11	0.87
100–200	0.07	0.91	−0.04	1.34	0.17	1.00	0.23	1.01	0.40	1.15	0.44	1.22
>200	0.01	1.03	−0.10	1.06	0.19	1.06	−0.13	1.02	−0.08	1.00	−0.06	1.05
	*F*	*p*	*F*	*p*	*F*	*p*	*F*	*p*	*F*	*p*	*F*	*p*
	0.102	0.9587	1.873	0.1341	5.371	0.0013	5.432	0.0012	2.371	0.0705	5.163	0.0017

**Table 4 ijerph-20-00978-t004:** Correlations between digital and e-Health variables, depending on using devices in physicians’ private and professional life.

In Private Life										
	Computer		Tablet		Smartphone		E-Mail		Mobile Apps	
	*F*	*p*	*F*	*p*	*F*	*p*	*F*	*p*	*F*	*p*
Digital literacy—own skills	9.480	0.0000	7.406	0.0001	3.506	0.0158	6.276	0.0004	8.420	0.0000
Digital literacy—the need for training	9.817	0.0000	1.580	0.1941	27.609	0.0000	14.514	0.0000	21.495	0.0000
The impact of the Internet/new technologies on healthcare and modern life	8.357	0.0000	9.068	0.0000	10.420	0.0000	11.179	0.0000	26.757	0.0000
Recommendation of e-Health solutions	9.241	0.0000	8.609	0.0000	6.229	0.0004	12.670	0.0000	10.941	0.0000
Evaluation of e-Health solutions—the patient	20.095	0.0000	16.166	0.0000	22.946	0.0000	43.822	0.0000	30.692	0.0000
Evaluation of e-Health solutions—medical facility	0.540	0.6550	2.606	0.0519	1.766	0.1536	5.141	0.0018	2.764	0.0421
At work										
Digital literacy—own skills	17.886	0.0000	11.169	0.0000	5.831	0.0007	19.753	0.0000	6.240	0.0004
Digital literacy—the need for training	7.165	0.0001	2.059	0.1057	10.942	0.0000	12.404	0.0000	16.157	0.0000
The impact of the Internet/new technologies on healthcare and modern life	19.639	0.0000	2.606	0.0519	11.731	0.0000	8.553	0.0000	6.962	0.0002
Recommendation of e-Health solutions	6.141	0.0005	3.878	0.0096	10.026	0.0000	12.827	0.0000	8.529	0.0000
Evaluation of e-Health solutions—the patient	26.764	0.0000	2.533	0.0571	16.946	0.0000	28.404	0.0000	10.060	0.0000
Evaluation of e-Health solutions—medical facility	2.396	0.0683	2.172	0.0913	0.491	0.6890	1.795	0.1481	0.875	0.4543

**Table 5 ijerph-20-00978-t005:** Correlations between: % of patients accessing online health information within a month, Opinion about the general quality of online health information, and Opinion on patients’ experience of health benefits, as a result of access to online health information and digital and e-Health variables.

	Digital Literacy—Own Skills		Digital literacy—The Need for Training		Assessment of the Impact of the Internet/New Technologies on Healthcare and Modern Life		Recommendation of e-Health Solutions		Evaluation of e-Health Solutions—The Patient		Evaluation of e-Health Solutions—Medical Facility	
	Mean	SD	Mean	SD	Mean	SD	Mean	SD	Mean	SD	Mean	SD
% of patients accessing online health information within a month												
<1%	−0.08	1.62	0.55	1.20	−0.31	0.83	−0.48	1.28	−0.38	0.69	−0.04	0.59
1–2%	−0.02	1.15	0.03	1.05	0.12	0.72	−0.15	1.51	0.48	1.30	−0.01	1.34
3–5%	−0.10	1.08	−0.15	0.70	−0.06	0.87	−0.01	0.95	−0.07	0.76	0.15	0.88
6–10%	0.03	0.89	−0.04	1.02	0.12	0.99	0.12	0.99	0.15	1.18	0.11	1.32
>10%	−0.14	0.70	0.27	0.91	−0.18	1.02	−0.07	0.84	−0.31	0.76	−0.09	0.77
it is difficult to estimate	0.25	1.15	−0.32	1.12	0.22	1.11	0.14	1.03	0.30	1.13	−0.03	1.11
Opinion about the general quality of online health information												
reliable	−1.00	1.07	0.06	0.93	−0.82	0.43	−1.52	1.15	−0.85	0.44	−0.16	0.76
usually reliable	−0.08	1.08	−0.28	0.87	−0.04	0.72	−0.29	0.89	−0.14	0.79	0.01	0.91
sometimes reliable	−0.07	0.83	0.10	0.95	−0.05	1.04	0.04	0.98	−0.07	0.79	0.15	1.03
unreliable	0.19	1.10	0.13	1.22	0.13	0.97	0.18	0.92	0.19	1.30	−0.23	0.90
no opinion	0.74	1.18	−0.44	0.94	0.54	1.31	0.67	0.68	0.75	1.59	−0.53	1.11
Opinion on patients’ experience of health benefits as a result of access to online health information												
often	−1.06	0.83	0.18	0.86	−0.50	0.57	−1.53	0.86	−0.84	0.39	−0.23	0.81
sometimes	0.00	0.64	0.04	0.98	−0.16	0.86	−0.30	0.79	−0.27	0.72	−0.05	0.94
seldom	−0.14	0.93	0.13	1.13	0.07	1.15	0.13	0.95	−0.06	1.08	0.03	1.05
never	0.65	1.10	0.98	1.08	0.57	0.89	1.05	0.59	0.73	1.14	−0.27	0.84
no opinion	0.19	1.23	−0.29	0.82	0.12	1.04	0.34	0.94	0.37	1.06	0.10	1.08
An assessment of patients’ use of online health knowledge												
very positive	−0.62	1.30	−0.10	0.99	−0.52	0.89	−1.51	1.14	−0.72	0.59	−0.11	0.77
positive	0.09	1.00	−0.09	0.66	0.13	0.95	−0.23	0.91	−0.12	0.66	−0.21	0.70
meaningless	−0.24	0.99	−0.01	1.08	−0.31	0.98	−0.21	1.10	−0.20	1.06	−0.17	0.97
negative	0.02	0.92	0.21	1.03	−0.01	0.82	0.19	0.85	0.04	0.96	0.21	1.08
very negative	0.31	1.07	−0.44	1.01	0.42	1.35	0.34	0.96	0.46	1.23	−0.05	1.07

**Table 6 ijerph-20-00978-t006:** Correlation matrix of selected variables.

	% of Patients Accessing Online Health Information within a Month		An Opinion on the Overall Quality of Online Health Information		An Opinion on Patients’ Health Benefits Related to Online Health Information		An Assessment of Patients’ Use of Online Health Knowledge	
	*F*	*p*	*F*	*p*	*F*	*p*	*F*	*p*
Digital literacy—own skills	1.541	0.1769	6.709	0.0000	7.845	0.0000	3.167	0.0143
Digital literacy—the need for training	4.262	0.0009	2.781	0.0270	6.066	0.0001	3.985	0.0036
The impact of the Internet/new technologies on healthcare and modern life	1.891	0.0956	3.776	0.0052	3.309	0.0113	4.754	0.0010
Recommendation of e-Health solutions	1.173	0.3225	11.079	0.0000	24.239	0.0000	10.246	0.0000
Assessment of e-Health solutions—the patient	5.193	0.0001	6.130	0.0001	11.394	0.0000	4.726	0.0010
Assessment of e-Health solutions—medical facility	0.517	0.7635	3.358	0.0104	0.760	0.5520	2.546	0.0396

**Table 7 ijerph-20-00978-t007:** Types of E-Health solutions recommended by physicians.

e-Health Solution	I Have Already Recommended		I Would Recommend		I Do Not Recommend	
	N	%	N	%	N	%
Remote monitoring of basic parameters (pressure, heart rate, temperature, glucose level)	45	14.7	127	41.4	135	44.0
Obtaining laboratory test results via the Internet	99	32.2	163	53.1	45	14.7
Arranging medical appointments via the Internet	83	27.0	143	46.6	81	26.4
Using a mobile application—the analysis of tests results	47	15.3	110	35.8	150	48.9
Using the mobile application—a knowledge base on health-related topics	41	13.4	151	49.2	115	37.5
Using the mobile application—a mobile drug database	48	15.6	146	47.6	113	36.8
Using a mobile application—reminder to take medication	41	13.4	222	72.3	44	14.3
Using a video consultation with the doctor /nurse /midwife to support the treatment process	33	10.7	132	43.0	142	46.3

**Table 8 ijerph-20-00978-t008:** Physicians’ opinion about selected e-Health solutions.

	Very Important		Important		Not very Important		Insignificant		No Opinion	
	N	%	N	%	N	%	N	%	N	%
Quick and easy access to the patient’s medical records in electronic form	129	42.0	127	41.4	31	10.1	6	2.0	14	4.6
The possibility to write electronic prescriptions	112	36.5	132	43.0	33	10.7	6	2.0	24	7.8
The possibility to write out electronic sick leaves	137	44.6	132	43.0	18	5.9	11	3.6	9	2.9
The possibility to write electronic referrals	112	36.5	141	45.9	31	10.1	8	2.6	15	4.9
Using the electronic database of medicines	148	48.2	128	41.7	13	4.2	2	0.7	16	5.2
The ability to remotely route patients to other specialists or hospitals	106	34.5	129	42.0	38	12.4	9	2.9	25	8.1
The solutions to streamline the sending/sharing clinical results	143	46.6	113	36.8	24	7.8	8	2.6	19	6.2
Solutions enabling remote patient care	87	28.3	84	27.4	44	14.3	45	14.7	47	15.3
More digital solutions supporting the treatment and self-monitoring of the patient’s health	70	22.8	143	46.6	46	15.0	13	4.2	35	11.4
The possibility to exercise comprehensive control over facilities, tracking generated costs, managing staff (schedules, schedules)	72	23.5	131	42.7	46	15.0	15	4.9	43	14.0
The possibility to conduct scientific research	96	31.3	120	39.1	26	8.5	18	5.9	47	15.3

**Table 9 ijerph-20-00978-t009:** Correlation matrix of all digital and e-Health indicators.

		Digital Literacy—Need for Training	The Impact of the Internet/New Technologies on Healthcare and Modern Life	Recommendation of e-Health Solutions	Assessment of e-Health Solutions—the Patient	Assessment of e-Health Solutions—Medical Facility
Digital literacy—own skills	*r*	0.000	0.447	0.413	0.449	−0.105
	*p*	1.0000	0.0000	0.0000	0.0000	0.0673
Digital literacy—need for training	*r*		−0.239	−0.131	−0.250	0.040
	*p*		0.0000	0.0220	0.0000	0.4883
The impact of the Internet/new technologies on healthcare and modern life	*r*			0.369	0.457	0.022
	*p*			0.0000	0.0000	0.7029
Recommendation of e-Health solutions	*r*				0.588	0.140
	*p*				0.0000	0.0144
Assessment of e-Health solutions—the patient	*r*					0.000
	*p*					1.0000

*r*—Pearson correlation coefficient.

## Data Availability

The data presented in this study are available on reasonable request from the corresponding author. The data are not publicly available due to restrictions, e.g., their containing information that could compromise the privacy of research participants.

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
