# Peer review of "Dr. Google: Physicians—The Web—Patients Triangle: Digital Skills and Attitudes towards e-Health Solutions among Physicians in South Eastern Poland—A Cross-Sectional Study in a Pre-COVID-19 Era"

_ijerph, 2023, doi:10.3390/ijerph20020978_

Round 1

Reviewer 1 Report (New Reviewer)

Thank you for the opportunity to review this manuscript, reporting on a cross-sectional survey to investigate physicians’ digital skill, and their opinions on obtaining health knowledge by patients, as well as physicians’ attitudes towards e-health solutions. This research is part of a larger study.

Strengths:

The research is well described, outlining the significance of the issue within the broader literature. Relevant theoretical and scholarly literature is integrated throughout. The methods for conducting the study and interpretation of results are described in detail, and supplementary data files provided additional details for understanding the results. The Conclusions are justified and supported by the results.

Limitations: Several editorial and style issues were identified—see below:

Title: Digital Skills, Perceptions of Dr. Google, and Attitudes of e-Health Solutions among Polish Physicians. A Cross-Sectional 3 Survey Study. Suggest changing ‘Attitude of” to Attitudes toward”

Abstract: line 19. Grammar: change rate to rated

Abstract: line 29. Change lower to lowered.

Line 37: Add ‘the’ before the ‘world’

Line 38: One of the most popular (sources).

Lines 67-75: Consider revising the sentence structure for clarity.

Lines 150-168: good discussion of TAM/UTAUT, please elaborate how this model has been applied/not applied in your design of the study or the interpretation of the results.

Line 474-477: is this statement complete, please check for missing text.

Line 499: “physicians questioned by Murray et al. in 85% claimed…  take out “in”

Line 510: Murray et al. reported that physicians did have a impression… change ‘a’ to ‘an’. Also change (challenging) to (challenged)

Lines 520-524: consider revising the sentence structure for clarity.

Lines 532 – 534: Revise statement for clarity.

Line 571: I suggest changing “interviewed” to “surveyed”.

Author Response

Reviewer 2 Report (New Reviewer)

Reviewers observations  13th December 2022

Information  about digital skills and attitudes towards eHealth solutions among physicians is of prime importance. The authors have surveyed 307 professionally active physicians between December 2019 to April 2020 in  South-Eastern Poland. The  majority of participants were employees of primary healthcare (81.4%)., Extrapolating observations from a convenience sample , restricted to a specific region and drawing conclusions for the whole country may have limitations. The reader would like to know if the age, sex, specialists distribution in the sample studied truly reflects their distribution in the physician population. The authors need to discuss more on the relevance of their sample size and heterogenity. Most important the study was conducted

before the  COVID-19 pandemic ( though acknowledged as a limitation) . It is universally accepted and proved that deployment in 2023 of eHealth solutions has radically changed as has the attitudes of physicians. If the study could be repeated now and compared with the original analysed data the value of the paper would considerably be increased. The title of the communication needs to be more specific and focused so that the reader is not misled eg Digital Skills and attitudes towards eHealth solutions among  physicians in South Eastern Poland in the pre Covid era  

Author Response

This manuscript is a resubmission of an earlier submission. The following is a list of the peer review reports and author responses from that submission.

Round 1

Reviewer 1 Report

The research gap in the literature is not very well discussed. The relevant literature, for example technology acceptance model (TAM), theory of planned behavior (TPB) etc., are not mentioned. There lacks good justification how this research contributes to the existing literature. Saying “To our knowledge, no research exists examining digital literacy, perception of online health information, and attitudes towards e-Health solutions from physicians' perspective in Poland” (page 5, line 143) is not enough to show the theoretical contribution.

Since this study consists of data in a pre-COVID-19 era, I would suggest the authors collect post-COVID-19 data, and make comparison. That would make this study more interesting.

The Cronbach's alpha values for “digital literacy” and “the impact of the Internet” are below 0.7 threshold. Authors need to check.